# Factors associated with delayed diagnosis of appendicitis in adults: A single-center, retrospective, observational study

Taku Harada[1,2]*, Yukinori Harada[2], Juichi Hiroshige[1], Taro Shimizu[2]

**1** Division of General Medicine, Showa University Koto Toyosu Hospital, Tokyo, Japan, **2** Department of Diagnostic and Generalist Medicine, Dokkyo Medical University Hospital, Mibu, Tochigi, Japan

* hrdtaku@gmail.com

**Data Availability Statement:** The datasets generated during and/or analyzed during the current study are available from the corresponding author on reasonable request. Because no approval

## Abstract

Appendicitis is one of the most common causes of acute abdominal pain; yet the risk of delayed diagnosis remains despite recent advances in abdominal imaging. Understanding the factors associated with delayed diagnosis can lower the risk of diagnostic errors for acute appendicitis. These factors, including physicians' specialty as a generalist or non-generalist, were evaluated through a retrospective, observational study of adult acute appendicitis cases at a single center, between April 1, 2014, and March 31, 2021. The main outcome was timely diagnosis, defined as "diagnosis at the first visit if the facility had computed tomography (CT) capability" or "referral to an appropriate medical institution promptly after the first visit for a facility without CT capability," with all other cases defined as delayed diagnosis. The frequency of delayed diagnosis was calculated and associated factors evaluated through multivariate and exploratory analyses. The overall rate of delayed diagnosis was 26.2% (200/763 cases). Multivariate analysis showed that tenderness in the right lower abdominal region, absence of diarrhea, a consultation of ≤6 h after symptom onset, and consultation with a generalist were associated with a decreased risk of delayed diagnosis of acute appendicitis. Exploratory analysis found that generalists performed more physical findings related to acute appendicitis, suggesting that this diagnostic approach may be associated with timely diagnosis. Future studies should adjust for other potential confounding factors, including patient complexity, consultation environment, number of physicians, diagnostic modality, and physician specialties.

## Introduction

Appendicitis is one of the most common causes of acute abdominal pain, with a lifetime risk of 8.6% in men, 6.7% in women [1], and an incidence rate of approximately 100–150 per 100,000 person-years [2]. However, the diagnosis of acute appendicitis is occasionally difficult and, despite recent advances in abdominal imaging studies, its misdiagnosis incidence rate has not decreased [3, 4]. Diagnostic errors are reported in 5.9–25.7% of adult acute appendicitis cases [5–8]. The misdiagnosis of acute appendicitis is associated with perforation,

has been obtained from the Ethics Committee for the sharing of de-identified data sets. This study was approved by the Ethics Committee of Showa University (21-096-A). Ethics Committee of Showa University 's mail address is m-rinri@ofc.showa-u.ac.jp.

**Funding:** T.H was supported by a grant-in-aid at this study from Japan Primary Care Association [grant number 04-02-001]. Y.H received Grants-in-Aid for Scientific Research from Japan Society for the Promotion of Science. Y.H received a payment for manuscript writing from PRECISION, Inc. S.T received Grants-in-Aid for Scientific Research from Japan Society for the Promotion of Science. S.T received a payment for manuscript writing from PRECISION, Inc. The funders had no role in study design, data collection and analysis, decision to publish, or preparation of the manuscript.

**Competing interests:** The authors have declared that no competing interests exist. This does not alter to PLOS ONE policies on sharing data and materials.

postoperative complications (persistent ileus, wound abscess, persistent drainage, and wound dehiscence), and increased length of hospital stay [9]. As 80% of the diagnostic errors are preventable [10], identifying the factors associated with misdiagnosis is necessary.

Previous studies have identified the following factors as being associated with diagnostic errors of acute appendicitis in adults: age >60 years [11, 12], appendicitis in the pelvic region [13], atypical symptoms or inadequate examination [14], poor clinical findings [5], the experience level of the emergency physician [15], female sex, the presence of coexisting conditions, constipation, and appendicitis without pain [8]. In addition, other physician- and environmental-related factors, such as physician specialty [16, 17], access to ultrasound [18], and hospital size [19], have been reported as factors associated with misdiagnosis in pediatric studies. However, no study has examined whether physician specialty causes the delayed diagnosis of acute appendicitis in adults.

In Japan, specialty training has historically been emphasized; therefore, both specialists and generalists provide assessment and treatment in primary and emergency care [20, 21]. A multicenter study in Japan regarding the delayed diagnosis of lower gastrointestinal perforation highlighted the possible effect of physician training on diagnostic accuracy for acute abdomen, with a lower rate of delayed diagnosis identified for generalists (primary care and emergency physicians) than for non-generalists [21]. However, the study did not address in detail the factors or processes that led generalists to make fewer diagnostic errors for lower gastrointestinal perforation [21]. Acute appendicitis, like lower gastrointestinal perforation, is an acute abdominal condition with a similar clinical presentation. Therefore, one would expect a lower rate of diagnostic delay in acute appendicitis by generalists, although, no studies have reported this to date.

Since there is no single clinically significant physical finding that is conclusive for the diagnosis of appendicitis [22] and there is no definitive laboratory test [23], clinical judgment, based on history taking and physical examination, is key in diagnosing acute appendicitis. A common pitfall in the diagnosis of appendicitis is including or excluding the diagnosis based on a single clinical sign or symptom [23]. Reports of physical findings made by attending physicians for general medicine inpatients leading to important changes in clinical management [24] and reports that inadequate physical examination causes preventable medical errors [25] suggest that improved accuracy of physical examination findings is essential to an excellent diagnostic process for acute appendicitis. While there have been studies on the degree of agreement among physicians for the same physical examination performed, differences in whether or not physicians perform each physical examination (or alternatively, whether or not they record findings) would be worth exploring. As an example, while rectal examination has been reported to not contribute much to the diagnostic accuracy of appendicitis [26], Rusnak et al. reported fewer diagnostic errors for appendicitis cases with documented rectal examination [14]. This suggests that the physician's attitude toward physical examination, rather than a direct effect of physical examination, may be related to the accuracy of appendicitis diagnosis.

To address this gap in knowledge regarding the physical examination approach and diagnostic accuracy for acute appendicitis in adults, we conducted a retrospective study of acute appendicitis cases, with the diagnosis confirmed by computed tomography (CT), to determine whether physician specialty is a potential source of delayed acute appendicitis diagnosis. Additionally, we hypothesized that consultation with a generalist, especially with respect to physical examination, would lower the incidence of delayed acute appendicitis diagnosis. The most vital component for a timely diagnosis of acute appendicitis is suspecting the disease based on the history taking and physical examination, which can be conducted irrespective to the location. Therefore, we included patients who directly visited our hospital, as well as patients who were referred from other clinics.

## Materials and methods

### Study design, setting, and statement of ethics

This was a retrospective observational analysis of patients diagnosed with acute appendicitis at Showa University Koto Toyosu Hospital, between April 1, 2014, and March 31, 2021. Showa University Koto Toyosu Hospital is a 400-bed acute care hospital in Tokyo. At Showa University Koto Toyosu Hospital, patients with abdominal pain in the emergency room are evaluated by physicians from various departments, including general internal medicine, gastroenterology, and surgery, depending on the situation at the time. In addition, in most Tokyo clinics, physicians from various departments provide primary care. They also evaluate patients with abdominal pain and refer them to a nearby acute care hospital if needed.

Our study followed the STROBE reporting statement for observational studies and the protocol was approved by the Ethics Committee of Showa University (21-096-A). An opt-out method was used so that patients could refuse to participate in the study.

### Study group

We used two datasets from Showa University Koto Toyosu Hospital to extract cases of appendicitis: the Japanese Healthcare Insurance system database and the radiology reports of CT. In this study, we extracted the medical records with the disease code of appendicitis in the insurance system database or with the diagnosis of appendicitis documented in the CT report. Eligible patients were ≥ 15 years of age who came directly to our hospital or were referred to our hospital from other clinics, between April 1, 2014, and March 31, 2021. In the referred groups, simple blood tests, such as complete blood count, was conducted on some patients. Medical records of eligible patients were reviewed to identify patients whose radiology reports described appendicitis as the diagnosis. Only the first event was included for patients with repeat visits for possible acute appendicitis to avoid the confounding effects of prior notes on the diagnostic process. Excluded were cases in which the final diagnosis was not acute appendicitis, the onset of appendicitis was during the period of hospitalization, the diagnosis was stump appendicitis, and prior hospitalization for acute appendicitis at another hospital, with patients transferred to our hospital for treatment.

### Data collection and definition of variables

In this study, the diagnosis of acute appendicitis was defined as the diagnosis in the radiology CT report. Patient factors (age, sex, the presence of underlying disease, consultation >6 h after symptom onset [27], and no history of appendicitis), disease factors (fever, constipation, diarrhea, absence of pain, migration of pain, anorexia, nausea and vomiting, recurrent pain, right lower quadrant tenderness, body temperature, white blood cell count, [WBC], WBC fraction, C-reactive protein [CRP] level, and location of the appendix), and environmental factors (specialty of the consulting physician and whether the initial medical examination was performed at a clinic or hospital) were retrospectively collected. The Charlson Comorbidity Index was used instead of the Elixhauser Comorbidity Index owing to its limitations in a retrospective study design [8, 28]. The cutoff values related to WBC in the Alvarado score were used as the criterion of a shift to the left for the WBC and neutrophil counts [29]. The cutoff value for elevated CRP, a frequently used inflammation marker in daily practice, was 10 mg/L [30]. CT was used in all cases to confirm the location of the appendix and to assess whether the appendix had extended toward the pelvic region. Symptoms and physical findings were confirmed using physicians' and nurses' medical records at Showa University Koto Toyosu Hospital and other records, such as scanned referral letters, medical questionnaires, and imaging data.

Information on patients visited by referral was collected first from the information in the patient referral document, and the missing information was made up with the information at Showa University Koto Toyosu Hospital. Physician specialty was identified from medical records, referral forms, and internet searches. Board-certified primary care and emergency physicians and the trainees were classified as generalists, as previously described [21]; other board-certified physicians and the trainees were classified as non-generalists, with subclassifications of gastroenterologists and non-gastroenterologists. Symptoms of acute appendicitis were determined by new onset symptoms among those listed under "disease factors." A medical record review was performed solely by the corresponding author (TH).

For the exploratory analyses, we included only patients who presented directly to Showa University Koto Toyosu Hospital as their first medical institution after symptom onset. Detailed information was collected via a review of medical records. As a result, the following information on physical findings was extracted: right lower abdominal tenderness, tenderness at McBurney's point, tenderness at Lanz's point, cough, percussion, heel drop, psoas, and obturator test. Right lower abdominal tenderness, tenderness at McBurney's point, and tenderness at Lanz's point were physical findings related to the right lower abdomen; and the cough [31–36], percussion [31–33, 35, 36], and heel drop tests [37–39] were physical tests aimed at detecting minor perineal irritation. The psoas and obturator tests were classified as tests that could detect pelvic appendicitis [36, 40].

## Main outcomes

The main outcome of our study was the frequency of delayed diagnosis of acute appendicitis and the associated factors. We defined two criteria for timely diagnosis, as previously described [10]: "diagnosis at the first visit if the facility had CT capability" and "referral to an appropriate medical institution promptly after the first visit for a facility without CT capability." In other words, CT was performed at the first visit to Showa University Koto Toyosu Hospital in almost all patients in the timely diagnosis group. All other cases were defined as delayed diagnoses and considered "lost opportunities" for timely diagnosis of acute appendicitis.

## Statistical analysis

Differences in nominal variables between the timely and delayed diagnosis groups were evaluated using the chi-square or Fisher's exact test, with a t-test or Wilcoxon rank-sum test used for continuous variables, as appropriate for the data distribution. Multivariate logistic regression analysis for delayed diagnosis (diagnostic error) considered the following factors suggested by existing literature: female sex, constipation, no pain at presentation, presence of comorbidities, pelvic appendicitis, and age >60 years. In addition, the following factors, which may be relevant to the diagnostic process and diagnostic errors were also included: CRP ≥10 mg/L, diarrhea, no tenderness in the right lower quadrant, history of appendicitis, a consultation >6 h after symptom onset, and physician training (generalist or non-generalist). We also conducted a sensitivity analysis for multivariate logistic regression analysis by using multiple imputation method to handle missing data. We used the "mice" package of R for multiple imputation with 20 imputations.

The sample size was targeted at approximately 1000 cases based on a predicted 20% incidence of delayed diagnosis of acute appendicitis and a planned 12-item multivariate analysis.

Statistical analyses were performed using the R 4.1.0 (The R Foundation for Statistical Computing, Vienna, Austria). All tests were two-tailed, with a *P* value of <0.05 considered statistically significant.

### Exploratory analysis

To clarify the effects of the difference in variety of physical examinations performed by generalists and non-generalists on the delayed diagnosis of appendicitis, we also conducted exploratory analyses. For the exploratory analyses, we included only patients who presented directly to Showa University Koto Toyosu Hospital as their first medical institution after symptom onset to avoid biases introduced by information from other medical institutions.

## Results

A total of 2180 eligible cases were identified: 1428 from our hospital's disease registry and 752 from radiology reports. Of these eligible cases, 1417 were excluded for the following reasons: duplicate cases (n = 855), repeat consultation (n = 26), non-acute appendicitis (n = 513), appendicitis developing during hospitalization (n = 4), stump appendicitis (n = 1), and transfers to our hospital for treatment after diagnosis at another center (n = 18). After screening, 763 cases were included in our analysis (Fig 1). Patients in our sample group were 40.9±15.4 years of age, and 330 (43.3%) were women.

Delayed diagnoses occurred in 200/763 cases, for an overall incidence rate of 26.2%. Delayed diagnoses were identified in 89 of 339 (26.3%) cases assessed by a gastroenterologist, in 76 of 299 (25.4%) cases assessed by a non-gastroenterologist, and in 3 of 64 (4.7%) cases assessed by a generalist. The incidence of delayed diagnosis was significantly lower for generalists than for gastroenterologists or non-gastroenterologists ($P<0.001$).

A comparison of patient characteristics between the delayed and timely diagnosis groups is presented in Table 1. On multivariate analysis, the following factors were associated with delayed diagnosis (Table 2): female sex (odds ratio [OR]: 1.95; 95% confidence interval [CI]: 1.21–3.16), no tenderness in the right lower abdominal region (OR: 7.32; 95% CI: 3.45–16.2), diarrhea (OR: 1.91; 95% CI: 1.09–3.34), a consultation of >6 h after symptom onset (OR: 2.43; 95% CI: 1.16–5.37), and consultation with a non-generalist (OR: 16.8; 95% CI: 3.19–315). Although age was analyzed as a continuous variable, it was not a significant factor in delaying the diagnosis of acute appendicitis in this study. Regarding the variables included in the multivariate analysis, at least one missing data existed in 305/763 cases (40.0%): 253 in constipation (33.2%), 184 in diarrhea (24.1%), 61 in physician training (8.0%), 22 in no tenderness in the right lower abdominal region (2.9%), 8 in no pain at presentation (1.1%), 7 in pelvic appendicitis (0.9%), 5 in a consultation >6 h after symptom onset (0.7%), and 2 in CRP $\geq$10 mg/L (0.3%). The prevalence of at least one missing data was not different between the delayed diagnosis (218/563, 38.7%) and timely diagnosis (87/200, 43.5%) group (P = 0.24). In the multivariate analysis with multiple imputations, no tenderness in the right lower quadrant (OR: 5.93; 95% CI: 3.11–1.3), diarrhea (OR: 1.96; 95% CI: 1.21–3.18), consultation of >6 h after symptom onset (OR 2.38; 95% CI: 1.27–4.45), and consultation by a non-generalist (OR 7.00; 95% CI: 1.97–24.9) remained significant risk factors for delayed diagnosis.

The exploratory analysis identified delayed diagnoses in 31 of the 247 cases, with a rate of 14.6% (30/206 cases) for non-generalists and 2.5% (1/40 cases) for generalists. Therefore, the rate of delayed diagnosis was significantly lower for generalists than for non-generalists ($P<0.001$).

Physical findings in the right lower abdomen were described by non-generalists in 83.1% (172/207) and by generalists in 90% (36/40) of their cases. Physical findings of mild peritoneal irritation were described by non-generalists in 31.9% (66/207) and by generalists in 82.5% (33/40) of their cases. Lastly, physical examination findings associated with pelvic appendicitis were described by non-generalists in 4.8% (10/207) and by generalists in 37.5% (15/40) of their cases. Generalists were significantly more likely to perform physical tests to detect mild

```
┌─────────────────────────────────────────────────────────┐
│ 2,180 events                                              │
│   1,428   events by registry cases                        │
│   752     events by radiologist's report                  │
└─────────────────────────────────────────────────────────┘
          │
          │        ┌────────────────────────────────────────────────────┐
          │        │ Exclusion                                           │
          │        │                                                      │
          │───────▶│ Duplicate case (n=855)                              │
          │        │ Not acute appendicitis (n=513)                      │
          │        │ Presented multiple times (n=26)                     │
          │        │ Developed during hospitalization (n=4)              │
          │        │ Stump appendicitis (n=1)                            │
          │        │ Transfer for treatment (n=18)                       │
          │        └────────────────────────────────────────────────────┘
          ▼
┌──────────────────────┐
│ 763 patients         │
└──────────────────────┘
          │
          ▼
┌──────────────────────┐
│ Exploration analysis │
│ 247 patients         │
└──────────────────────┘
```

**Fig 1. Flow chart of patient selection.**

**Table 1. Comparison of patient characteristics and factors associated with "delayed" and "timely" diagnosis.**

| Characteristics | Delayed Diagnosis Group (n = 200) Frequency (%) | Timely Diagnosis Group (n = 563) Frequency (%) | P value |
|---|---|---|---|
| Sex (Female) | 96/200 (48.0) | 234/563 (41.6) | 0.12 |
| Age (years, mean ± SD) | 40.7 ± 14.5 | 41.0 ± 15.7 | 0.79 |
| Age >60 years | 18/200 (9.0) | 67/563 (11.9) | 0.29 |
| Consultation >6 h after symptom onset | 177/199 (88.9) | 422/559 (75.5) | <0.001 |
| Diarrhea | 41/153 (26.8) | 65/426 (15.3) | 0.002 |
| Constipation | 21/146 (14.4) | 37/364 (10.2) | 0.18 |
| No pain | 3/197 (1.5) | 5/558 (0.9) | 0.43 |
| Migration of pain | 50/191 (26.2) | 161/552 (29.2) | 0.45 |
| Anorexia | 140/185 (75.7) | 406/522 (77.8) | 0.61 |
| Nausea and vomiting | 92/184 (50.0) | 259/525 (49.3) | 0.93 |
| Rebound tenderness | 95/188 (50.5) | 272/532 (51.1) | 0.93 |
| Negative tenderness in the right lower quadrant | 30/193 (15.5) | 34/548 (6.2) | <0.001 |
| Fever (body temperature ≥37.4°C) | 63/190 (33.2) | 146/546 (26.7) | 0.09 |
| White blood cell ≥10000/μL | 135/198 (68.2) | 418/653 (64.0) | 0.12 |
| Neutrophil left shift (≥75%) | 145/191 (75.9) | 426/549 (77.6) | 0.62 |
| CRP ≥10 mg/L | 148/198 (74.7) | 312/563 (55.4) | <0.001 |
| Presence of comorbidities | 11/200 (5.5) | 49/563 (8.7) | 0.17 |
| No history of appendicitis | 191/200 (95.5) | 491/563 (87.2) | 0.001 |
| Pelvic appendicitis | 109/196 (55.6) | 226/560 (40.4) | <0.001 |
| Non-gastroenterologist consultation at the first visit | 76/168 (45.2) | 223/534 (41.8) | 0.47 |
| Gastroenterologist consultation at the first visit | 89/168 (53.0) | 250/534 (46.8) | 0.18 |
| Generalist consultation on the first visit | 3/168 (1.8) | 61/534 (11.4) | <0.001 |
| The first consultation at a clinic | 131/200 (65.5) | 240/563 (42.6) | <0.001 |

CRP: C-reactive protein

peritoneal irritation and pelvic appendicitis than non-generalists (*P*<0.001). Overall, the positive rate of physical findings was not different between non-generalists and generalists: tenderness in the right lower abdomen, 75.0% (129/172) and 66.7% (24/36), respectively (*P* = 0.31);

**Table 2. Results of the multivariate logistic regression analysis to identify factors associated with a delayed diagnosis of acute appendicitis.**

| Multivariate logistic regression analysis | Odds ratio (95% CI) | P value |
|---|---|---|
| Sex (Female) | 1.95 (1.21, 3.16) | 0.01 |
| Constipation | 1.04 (0.47, 2.20) | 0.91 |
| Absence of pain | 0.99 (0.10, 7.20) | >0.99 |
| Patients with comorbidities | 0.37 (0.11, 1.04) | 0.08 |
| Pelvic appendicitis | 1.13 (0.70, 1.82) | 0.62 |
| Age (for one year increase) | 0.99 (0.98, 1.01) | 0.29 |
| CRP ≥10 mg/L | 1.78 (1.00, 3.27) | 0.06 |
| No tenderness in the right lower quadrant | 7.32 (3.45, 16.2) | <0.001 |
| Diarrhea | 1.91 (1.09, 3.34) | 0.02 |
| No history of appendicitis | 2.60 (0.92, 9.52) | 0.10 |
| Consultation >6 h after symptom onset | 2.43 (1.16, 5.37) | 0.02 |
| Consultation by a non-generalist | 16.8 (3.19, 315) | 0.01 |

mild peritoneal irritation, 74.2% (49/66) and 72.7% (24/33), respectively (*P* = 1.00); and pelvic appendicitis, 50.0% (5/10) and 46.7% (7/15), respectively (*P* = 1.00).

## Discussion

We identified a 26.2% overall rate of delayed acute appendicitis diagnosis. We also identified female sex, the absence of right lower abdominal pain, initial consultation at a clinic, and a consultation >6 h after symptom onset as factors associated with a delayed diagnosis. Assessments performed by non-generalists increased the likelihood of delayed diagnosis, with a rate of 25.8% (165/638 cases) for non-generalists compared to 4.7% (3/64 cases) for generalists. After multivariate analysis and adjustment for other factors, consultation with a non-generalist remained a risk factor for delayed diagnosis. Our exploratory analysis identified that generalists were more likely than non-generalists to examine patients for signs of peritoneal irritation and pelvic appendicitis. However, because this was based on exploratory analysis, causation between the diagnostic process of generalists and a lower likelihood of delayed diagnosis cannot be inferred.

The frequency of delayed diagnosis in our study was higher than that reported in previous studies [5–8]. Previous reports of diagnostic errors in appendicitis have mainly focused on negative appendectomy, which could not address the diagnostic errors in patients who were treated conservatively. Recently, Mahajan et al. reported an incidence rate of diagnostic error for appendicitis in adults of 6.0%; however, this rate of misdiagnosis may be underestimated for an epidemiologic study as it was based on the definition of "potential misdiagnosis of appendicitis" and used big data [8]. Another recently reported study by Leung et al. compared diagnosis on admission with diagnosis on discharge, including non-resected cases, and reported a diagnostic error rate for acute appendicitis of 19% [6]. On the other hand, a recent large cohort study in pediatrics reported a delay rate of <5% in the diagnosis of appendicitis [8, 41]. Michelson. et al. reported a 63% delay in appendicitis diagnosis in a study that included only patients in whom a revisit led to the appendicitis diagnosis [42]. This large difference in the frequency of diagnostic errors for appendicitis may be influenced by differences in the definition of diagnostic error, as well as differences in context, including clinical practice and differences in health care systems [43]. We believe that the frequency of delayed diagnosis in our study was higher than the rate reported in previous studies because we used strict the criteria for timely diagnosis to reduce bias, including only cases with a correct diagnosis made at the time of an initial visit as timely diagnosis. Our findings on female sex and the absence of right lower abdominal pain as risk factors of delayed diagnosis were consistent with those of previous studies [5, 8, 14]. Tenderness in the right lower abdomen was the most useful clinical sign of acute appendicitis, consistent with previous reports [22, 44], and the absence of this sign made diagnosis difficult.

A key finding in our study was that generalists were more likely than non-generalists to perform various physical tests. Generalists did not limit their assessments to the right lower abdomen or rely on tenderness in the right lower abdomen for diagnosis. Specifically, generalists may include tests for minor peritonitis (cough, percussion, and heel drop tests) [31–39] and appendicitis in the pelvic region (psoas and obturator test) to improve their diagnostic accuracy. Our finding that generalists may have a more complete diagnostic process than non-generalists was consistent with a previous study which reported that, compared to non-generalists, generalist physicians have a lower risk of delayed diagnosis for lower gastrointestinal perforation, another acute abdomen condition [21]. In recent years, it has been noted that interest in physical examination has been waning with advances in diagnostic imaging [45]. However, general medicine departments in Japan provide an effective clinical situation for educating

residents about physical examination [46, 47]. These backgrounds may be related to the present results, in which generalists recorded a greater variety of physical findings.

The strengths of our study include collecting information on physician specialty and diagnostic processes through a detailed review of medical records and evaluating the association between these factors and delayed diagnosis of acute appendicitis. The limitations of our study should also be acknowledged. First, this is a single-center study that used a retrospective design. Therefore, selection and information bias cannot be ruled out, nor external validity be examined. Second, objective criteria were used for CT findings and diagnostic delay but were evaluated by a single reviewer. Third, whether at the clinic or hospital, the number of physicians and outpatients and differences among patients may have been poorly adjusted for by the wide variety of confounding factors in different consultation settings. Fourth, the exploratory analysis regarding physical examination was univariate only and the sample size was small, making it difficult to adjust for other factors fully. Moreover, although the record of physical findings was verified because of the design of this study, it is impossible to know if a physician performed a particular physical examination but did not document it in the medical record. Of note, however, the fact that a physical examination finding is documented in the medical record indicates that the physician thought the finding was important to the diagnosis. Thus, the results suggest that the generalist was interested in a more detailed search for intra-abdominal disease. Fifth, only cases of acute appendicitis confirmed by abdominal CT were included; therefore, cases diagnosed by sonography, which included pregnant women, were excluded. CT scan is a highly sensitive and specific test for acute appendicitis. Of course, clinical diagnosis and ultrasonography are also important, but clinical judgment and ultrasonography are often not documented and there are differences in accuracy among examiners. Since this study focused on "diagnostic error," we included cases with confirmed appendicitis by CT as criteria, emphasizing information for which more objective data remained. In Japan, there are many CT imaging centers, with good accessibility [48]; therefore, CT is commonly used to make decisions in adult cases. However, the use of CT is limited by radiation exposure and differences in accessibility. It should be noted that our study was conducted in central Japan, where access to CT is good in both clinics and hospitals. Sixth, our study focused on the outcome of delayed diagnosis in cases in which acute appendicitis was confirmed by CT imaging. The use of other imaging modalities, prognosis, length of hospital stay, and complications were not investigated; therefore, it is impossible to determine whether early diagnosis by CT is the optimal diagnostic process. Also, it is possible that generalists are performing CT imaging more frequently than non-generalists. In addition, the use of contrast media in CT imaging is left to the discretion of each clinician and we did not investigate the association between contrast CT and delayed diagnosis. The difference in sensitivity of CT for acute appendicitis due to the use of contrast media is minimal, around 1%, and we did not collect information in this study because we judged the association with delayed diagnosis of appendicitis to be very low [48]. Finally, the sensitivity of CT for the diagnosis of acute appendicitis is not 100% [49, 50] and cases of missed diagnosis by CT were not investigated in this study.

## Conclusions

In patients who were diagnosed with appendicitis by the CT findings, a delayed diagnosis occurred in about one fourth of the cases. Consultation by generalists at the index visit was associated with a lower occurrence of delayed diagnosis. Our findings further suggest that generalists may include a wider range of physical examinations to detect peritoneal irritation, which may explain the lower rate of delayed diagnosis, compared to the other specialties. Future studies are warranted to clarify this finding by adjusting for other important confounding factors, such as patient complexity and clinical context.

## Acknowledgments

We thank Editage (www.editage.jp) for English language editing.

## Author Contributions

**Conceptualization:** Taku Harada, Yukinori Harada.

**Data curation:** Taku Harada.

**Formal analysis:** Taku Harada, Yukinori Harada.

**Funding acquisition:** Taku Harada, Yukinori Harada.

**Investigation:** Taku Harada.

**Methodology:** Taku Harada, Yukinori Harada.

**Supervision:** Yukinori Harada, Juichi Hiroshige, Taro Shimizu.

**Writing – original draft:** Taku Harada.

**Writing – review & editing:** Yukinori Harada, Juichi Hiroshige, Taro Shimizu.

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
