## [Decision Letter · Decision Letter 0]

8 Jul 2022

PONE-D-22-16651Factors associated with delayed diagnosis of appendicitis in adults: a single-center, retrospective, observational studyPLOS ONE

Dear Dr. Harada,

Thank you for submitting your manuscript to PLOS ONE. After careful consideration, we feel that it has merit but does not fully meet PLOS ONE’s publication criteria as it currently stands. Therefore, we invite you to submit a revised version of the manuscript that addresses the points raised during the review process.

We look forward to receiving your revised manuscript.

Kind regards,

Kenneth A Michelson, MD MPH

Academic Editor

PLOS ONE

Journal Requirements:

"T.H was supported by a grant-in-aid at this study from Japan Primary Care Association [grant number 04-02-001].

Y.H received Grants-in-Aid for Scientific Research from Japan Society for the Promotion of Science.

Y.H received a payment for manuscript writing from PRECISION, Inc.

S.T received Grants-in-Aid for Scientific Research from Japan Society for the Promotion of Science.

S.T received a payment for manuscript writing from PRECISION, Inc."

"T.H was supported by a grant-in-aid at this study from Japan Primary Care Association [grant number 04-02-001].

Y.H received Grants-in-Aid for Scientific Research from Japan Society for the Promotion of Science.

Y.H received a payment for manuscript writing from PRECISION, Inc.

S.T received Grants-in-Aid for Scientific Research from Japan Society for the Promotion of Science.

S.T received a payment for manuscript writing from PRECISION, Inc."

We note that you received funding from a commercial source: PRECISION, Inc.

Additional Editor Comments:

I have significant concerns about the design and some of the conclusions of the study:

- One of the central results is that generalists are less likely to have a delayed diagnosis due to differences in the physical exam. However, the medical record only tells you whether an exam maneuver was documented, not whether it was performed. How sure can you be that exam maneuvers that were not documented did not occur?

- The rate of delayed diagnosis (>25%) is dramatically higher than in other studies (<5% typically). This is a red flag and makes me question the study results. Is it possible that selection bias existed?

- There are not enough patients in the study to perform a regression with 19 variables. I strongly recommend statistical consultation and revision of the modeling plan.

Reviewers' comments:

Reviewer's Responses to Questions

**Comments to the Author**

1. Is the manuscript technically sound, and do the data support the conclusions?

Reviewer #1: Partly

Reviewer #2: No

2. Has the statistical analysis been performed appropriately and rigorously? 

Reviewer #1: I Don't Know

Reviewer #2: No

3. Have the authors made all data underlying the findings in their manuscript fully available?

Reviewer #1: No

Reviewer #2: No

4. Is the manuscript presented in an intelligible fashion and written in standard English?

Reviewer #1: Yes

Reviewer #2: No

5. Review Comments to the Author

Reviewer #1: Title: Factors associated with delayed diagnosis of appendicitis in adults: a single-center,

retrospective, observational study (PONE-D-22-16651)

Methodology: retrospective study

Authors’ main findings: In 26% (200/763) patients, diagnosis of appendicitis was delayed, defined as any possibility other than “diagnosis at the first visit if the facility had computed tomography (CT) capability” or “referral to an appropriate medical institution promptly after the first visit for a facility without CT capability”. The factors associated with dealyed dianosis following multivariate analysis were female sex, absence of right lower quadrant tenderness, first consultation at clinic, and consultation by a non-generalist.

Reviewer’s comments: The topic of timely diagnosis of acute appendicitis is an important topic for emergency physicians and general surgeons. I has many difficulties with this study.

1. The set up of the study is unclear – The study was carried out in one hospital in Tokyo, I assume with CT scan. It is unclear, who evaluates patients in the emergency department? The classification of physicians is also uncear since it includes specialists - who are these? emergency physicians? general surgeons?, others? It includes generalists – a term which is unknown to me, and non-generalists, another unknown term. The authors ought to explain the set up. How do the patients reach this hospital. Are they referred by their family doctors? Are they referred by their family doctors after some outside workup was done? As written above, who are these physicians who examine the patients in the emergency department (what is their training)?

2. The authors’ main endpoint relates to if CT was done or not done. Is CT compulsary for the diagnosis of appendicitis in the auhors’ institution? This is not common practice in many countries.

3. The authors’ finding was that generalists were more likely than non-generalists to examine patients for signs of peritoneal irritation and pelvic appendicitis. I understand that some would send more laboratory exams or imaging exams than others. I do not understand how it can be possible that one physician is more likely than another physician to examine the abdomen of a patient presenting to the emergency department complaining of abdominal pain.

4. Twenty variables were included in the multivariate regression analysis. It is unclear whether interactions were included as well. I would suggest a statistician to evaluate the analysis.

Reviewer #2: Comment to the authors：

The authors report that the present study reveals factors associated with delayed diagnosis of appendicitis. Particularly, they insist that generalists perform physical examinations well and are less likely to have delayed diagnoses of appendicitis. Although the details are very interesting, there is a certain dissociation between the purpose of the study and its conclusions. Additionally, the fatal flaw of this study is that the methodology for avoidance and adjustment for possible bias is not enough, though I understand that this is a retrospective study. Therefore, the authors can not insist on their current conclusions based on the findings of this study, and major revisions are required.

Major points

Introduction:

1. The authors provide adequate comments on the delayed diagnosis of appendicitis. However, there is a lack of information regarding the association between generalists and physical examination, and between appendicitis and physical examination. It is unclear contextually why the authors hypothesized that generalists (or physical examination) would lead to a lower delayed diagnosis of appendicitis. The authors should add information with references in the literature.

2. The objectives stated in the Introduction, and the conclusions of this study seem to be disconnected. The authors should conclude with discussions of the study's findings with objective information already known and a clear answer to the objectives and hypotheses.

Method：

3. Informed consent should not be waived for a retrospective study. At least comprehensive consent or opt-out should be implemented and mentioned in the main text.

4. Please clarify the symptoms of appendicitis or CT findings used for inclusion, objectively and specifically, with references if possible. Additionally, please clarify how the accuracy of the review of medical records was ensured, e.g., how many reviewers were responsible for the review.

5. The inclusion criteria are not fully explained.

6. When was the CT performed, and when did the patients have the reviewed symptoms?

7. Please explain how missing data is managed. Mainly, how the authors managed physical findings and symptoms being not recorded is essential. Please clarify whether the authors counted them as none or missing. Additionally, please describe in the limitation how this may affect the results of this study.

8. How were the collected items selected? Did the authors review the literature?

9. Please explain whether CT with or without contrast enhancement was considered and how and why it was determined.

10. Shouldn’t the diagnostic accuracies in other modalities be discussed?

11. I think that performing CT and referring patients for CT should be different. What is the meaning to analyze patients from clinics without CT and hospitals with CT as the same population? Please clarify in the main text.

12. The other biases that affect differences in the location of the first consultation seem not to be adjusted sufficiently.

13. Although the duration from onset (6 hours) and age (60 years) were analyzed as categorical variables, they must be analyzed as continuous variables. If the authors analyze them as categorical variables, they should explain how they set the cut-offs.

14. The bias for delayed diagnosis due to the clinical settings would not be adjusted unless the duration from the onset in detail is included in the covariate items.

15. The differences in the departments may reflect the differences in the visiting patients. Depending on the role of each department, the patients’ backgrounds should be different, such as a patient with a difficult diagnosis visiting a specialist or a first-time patient visiting a generalist. How did the authors address such biases?

16. What is the definition of the generalist in this study? Please clarify whether they are board-certified physicians or simply belong to a general medicine department.

17. Because the primary outcome of this study is a delayed diagnosis, potential confounding factors such as type of department, number of medical doctors, number of beds, and number of outpatients in each department should be clarified and adjusted in the multivariate analysis.

18. The first paragraph of the Exploration analysis should be included in the background.

19. The second paragraph of the Exploration analysis should be described in Data collection and definition of variables.

20. The number of patients should not be mentioned as 247 in the method. This should be mentioned in the results.

Discussion:

1st paragraph

21. This study design does not adequately adjust for differences between patients in specialist and generalist departments. It is impossible to determine whether the difference between the two groups is truly significant unless the biases such as patient characteristics, symptom course, and other conditions are adjusted.

3rd paragraph

22. This is somewhat of a leap in logic. While it is true that the generalist performed CT earlier, this study's results cannot determine if this was the correct diagnostic practice.

23. Since there are many factors to be considered, such as patients for whom CT was not performed, patients diagnosed by other modalities, and differences in patients according to the department, it would be preferable to use a non-judgmental expression here.

24. Additionally, the physical examination was only analyzed in univariate analysis, and it is difficult to adjust adequately with other factors considering the small sample size. I believe this is a critical flaw in this study to emphasize the superiority of the generalists' physical examination.

25. I cannot understand the relation of the 4th paragraph to the results or the conclusions of this study.

Conclusion:

26. Please revise the conclusion based on the hypothesis, objectives, findings, and appropriate discussions.

27. “Second, we could not verify whether physical findings, such as peritoneal irritation or pelvic inflammation, could prevent delayed diagnosis owing to the retrospective study design.” This section is not the conclusion that can be drawn from the findings of this study.

28. “Our findings will need validation in future multicenter and prospective studies.” I do not think it is necessary as a conclusion of this study.

6. PLOS authors have the option to publish the peer review history of their article (what does this mean?). If published, this will include your full peer review and any attached files.

Reviewer #1: No

Reviewer #2: No

---

## [Author Response · Author response to Decision Letter 0]

23 Aug 2022

Thank you very much for reviewing our manuscript and offering valuable advice.

We appreciate the invaluable comments that the reviewers provided, which we are confident have helped us improve the manuscript. We have provided point-by-point responses to each of the reviewers’ comments and describe the related revisions below. We have indicated changes via red characters in the revised manuscript.

Thank you again for your thorough and supportive peer review. 

On behalf of the co-authors,

Sincerely,

Taku Harada 

Division of General Medicine, Showa University Koto Toyosu Hospital, 

5-1-38 Toyosu Koto-ku, Tokyo, 135-8577, Japan

hrdtaku@gmail.com

■Additional Editor Comments:

Comment 1: One of the central results is that generalists are less likely to have a delayed diagnosis due to differences in the physical exam. However, the medical record only tells you whether an exam maneuver was documented, not whether it was performed. How sure can you be that exam maneuvers that were not documented did not occur?

Response 1: We agree with the editor that the medical record only shows whether an exam maneuver was documented, not whether it was performed. As this study was retrospective, we were not able to know that exam maneuvers that were not documented did not occur. Therefore, we described this issue as a limitation.

(P16 L314-321)

Fourth, the exploratory analysis regarding physical examination was univariate only and the sample size was small, making it difficult to adjust for other factors fully. Moreover, although the record of physical findings was checked because of the design of this study, it is impossible to know if the physician took a particular physical finding and did not document it in the medical record. However, the fact that a physical examination finding is documented in the medical record indicates that the physician thought the finding was important to the diagnosis. Thus, the results suggest that the generalist was interested in a more detailed search for intra-abdominal disease.

Comment 2: The rate of delayed diagnosis (>25%) is dramatically higher than in other studies (<5% typically). This is a red flag and makes me question the study results. Is it possible that selection bias existed? 

Response 2: The high rate of delayed diagnosis compared to other studies derived from the definition of the delayed diagnosis in our study. Previous reports of diagnostic errors in appendicitis have mainly focused on negative appendectomy, which could not address the diagnostic errors in patients who were treated conservatively. In addition, in our study, only cases that correct diagnosis was made at the time of initial visit were treated as timely diagnosis, which may also have increased the rate of delayed diagnosis. We addressed the issue in the discussion section as follows:

(P14 L275-P15 L290)

The frequency of delayed diagnosis in our study was slightly higher than that reported in previous studies [5‒8]. Previous reports of diagnostic errors in appendicitis have mainly focused on negative appendectomy, which could not address the diagnostic errors in patients who were treated conservatively. A recent report by Mahajan et al. found a 6.0% frequency of diagnostic error for appendicitis in adults, but this may be an underestimate for an epidemiologic study because it is based on the definition of "potential misdiagnosis of appendicitis" and uses big data [8]. Another recently reported study by Leung et al. compared diagnosis on admission with diagnosis on discharge, including non-resected cases, and reported a diagnostic error rate of 19% [6]. In addition, in our study, only cases that correct diagnosis was made at the time of initial visit were treated as timely diagnosis, which may also have increased the rate of delayed diagnosis. We believe that the higher frequency reflects our strict criteria for case selection, which excluded follow-up visits to reduce bias. Our findings on female sex and the absence of right lower abdominal pain as risk factors of delayed diagnosis were consistent with those of previous studies [5, 8, 14]. Tenderness in the right lower abdomen was the most useful clinical sign of acute appendicitis, consistent with previous reports [22, 42], and the absence of this sign made diagnosis difficult. 

Comment 3: There are not enough patients in the study to perform a regression with 19 variables. I strongly recommend statistical consultation and revision of the modeling plan.

Response 3: As the reviewer suggested, we revised the multivariate regression model. The new model included following 12 items: 7 items were selected based on the existing literature: female sex, constipation, no pain at presentation, presence of comorbidities, pelvic appendicitis, and age >60 years; and 5 items were selected that considered clinically important: CRP ≥10 mg/L, diarrhea, past history of appendicitis, a consultation >6 h after symptom onset, and physician training (generalist or non-generalist). Regarding the interaction terms, we did not include any interaction term in the model since we assumed that there was no potential factor that affected the diagnostic accuracy of physicians’ specialty. According to the changes, we revised the methods and results sections.

(P8 L179-L185)

“Multivariate logistic regression analysis for delayed diagnosis (diagnostic error) considered the following factors suggested by existing literature: female sex, constipation, no pain at presentation, presence of comorbidities, pelvic appendicitis, and age >60 years. In addition, the following factors, which may be relevant to the diagnostic process and diagnostic errors were also included: CRP ≥10 mg/L, diarrhea, no tenderness in the right lower quadrant, history of appendicitis, a consultation >6 h after symptom onset, and physician training (generalist or non-generalist).”

(P9 L189-191)

“The sample size was targeted at approximately 1000 cases, based on a predicted 20% incidence of delayed diagnosis of acute appendicitis and a planned 12-item multivariate analysis.”

(P10 L221-226)

“On multivariate analysis, the following factors were associated with delayed diagnosis (Table 2): female sex (odds ratio [OR]: 1.95; 95% confidence interval [CI]: 1.21–3.16), no tenderness in the right lower abdominal region (OR: 7.32; 95% CI: 3.45–16.2), diarrhea (OR: 1.91; 95% CI:1.09–3.34), a consultation of >6 h after symptom onset (OR: 2.43; 95% CI: 1.16–5.37), and consultation with a non-generalist (OR: 16.8; 95% CI: 3.19–315).”

■5. Review Comments to the Author

■Reviewer #1

Comment 1: The set up of the study is unclear – The study was carried out in one hospital in Tokyo, I assume with CT scan. It is unclear, who evaluates patients in the emergency department? The classification of physicians is also uncear since it includes specialists - who are these? emergency physicians? general surgeons?, others? It includes generalists – a term which is unknown to me, and non-generalists, another unknown term. The authors ought to explain the set up. How do the patients reach this hospital. Are they referred by their family doctors? Are they referred by their family doctors after some outside workup was done? As written above, who are these physicians who examine the patients in the emergency department (what is their training)?

Response 1: Thank you for your constructive comments. In many clinics in Tokyo, physicians from various specialties provide primary care in their clinics and treat patients with abdominal pain. At the hospital where this study was conducted, patients with abdominal pain are seen in the emergency room by physicians from emergency department, general internal medicine department, gastroenterology department, or surgery department, depending on the situation at the time. Regarding the specialty, we defined board-certified physicians of general medicine or emergency medicine and the trainees as generalists, and defined board-certified physicians of other than general medicine or emergency medicine and the trainees as non-generalists; and other physicians who were unknown for their board certification were treated as missing data for specialty. Consultation pattern of patients in this study were divided into two types: one was that patients directly visited the hospital, and the other was that patients referred from clinics to the hospital. We addressed these issues in the method section as follows:

(P5 L102-107)

“Showa University Koto Toyosu Hospital is a 400-bed acute care hospital in Tokyo. At Showa University Koto Toyosu Hospital, patients with abdominal pain in the emergency room are seen by physicians from various departments, including general internal medicine, gastroenterology, and surgery, depending on the situation at the time. In addition, in most Tokyo clinics, physicians from various departments provide primary care. They also see patients with abdominal pain and refer them to a nearby acute care hospital if needed.”

(P6 L117-L120)

Eligible patients were ≥ 15 years of age who came directly or were referred from other clinics to Showa University Koto Toyosu Hospital between April 1, 2014, and March 31, 2021. In the referred groups, simple blood tests such as complete blood count was conducted in some patients.

(P7 L149-153)

“Physician specialty was identified from medical records, referral forms, and internet searches. Including board-certified or training primary care and emergency physicians were classified as generalists, as previously described [21]; other board-certified physicians were classified as non-generalists, with subclassifications of gastroenterologists and non-gastroenterologists.”

Comment 2: The authors’ main endpoint relates to if CT was done or not done. Is CT compulsary for the diagnosis of appendicitis in the auhors’ institution? This is not common practice in many countries. 

Response 2: We agree with you that clinical and ultrasonographic diagnosis is important in acute appendicitis. However, clinical diagnosis and ultrasonography are often not well documented and there are differences in accuracy among examiners. Since this study is dedicated to "diagnostic errors", we focused on information for which more objective data is available and included cases in which appendicitis was confirmed by CT as a criterion. We modified the Discussion as follows:

(P16 L321-P17 L335)

Fifth, only cases of acute appendicitis confirmed by abdominal CT were included; therefore, cases diagnosed by sonography, which included pregnant women, were excluded. CT scan is a highly sensitive and specific test for acute appendicitis. Of course, clinical diagnosis and ultrasonography are also important, but clinical judgment and ultrasonography are often not documented, and there are differences in accuracy among examiners. Since this study focused on "diagnostic error," we included cases with confirmed appendicitis by CT as criteria, emphasizing information for which more objective data remained. Japan has many CT machines with good accessibility [46], and CT is commonly used to make decisions in adult cases. However, CT also has problems with radiation exposure and variation in accessibility. It should be noted that this study was conducted in central Japan, where access to CT is good in both clinics and hospitals. Sixth, this study focuses on the outcome of delayed diagnosis in cases in which a CT scan confirms the diagnosis of acute appendicitis. The use of other modalities, prognosis, length of hospital stays, and complications was not investigated; therefore, it is impossible to determine whether early diagnosis by CT is the optimal diagnostic process.

Comment 3: The authors’ finding was that generalists were more likely than non-generalists to examine patients for signs of peritoneal irritation and pelvic appendicitis. I understand that some would send more laboratory exams or imaging exams than others. I do not understand how it can be possible that one physician is more likely than another physician to examine the abdomen of a patient presenting to the emergency department complaining of abdominal pain.

Response 3: Some studies suggested that the contents of physical examination varied among physicians and may affect the diagnostic errors. For example, Rusnak et al. reported that the documentation of rectal examination was associated with fewer diagnostic errors of appendicitis, although no documentation is not the direct evidence that the exam was not conducted. Besides, in Japan, the curriculum for physicians in training of general medicine and emergency medicine underscores the importance of physical examination in diagnosis; whereas other specialty training mainly focuses on the diagnostic procedures such as endoscopy, ultrasonography, CT, or MRI. Such a difference of training may result in the difference of physical examination among physicians.

(P15 L299-304)

In recent years, it has been noted that interest in physical examination has been waning with advances in diagnostic imaging [43]. However, general medicine departments in Japan provide an effective clinical situation for educating residents about physical examination [44, 45]. These backgrounds may be related to the present results, in which generalists recorded a greater variety of physical findings.

(P4 L70-86)

“Since there is no single clinically significant physical finding that is conclusive for the diagnosis of appendicitis [22] and there is no definitive laboratory test, [23] clinical judgment based on history taking and physical examination is key in diagnosing acute appendicitis. A common pitfall in the diagnosis of appendicitis is including or excluding the diagnosis based on a single sign or symptom [23]. Reports of physical findings made by attending physicians on general medicine inpatients leading to important changes in clinical management [24] and reports that inadequate physical examination causes preventable medical errors [25] suggest that improved accuracy of physical examination findings is important for an excellent diagnostic process for acute appendicitis. While there have been studies on the degree of agreement among physicians when they take the same physical examination, there have been few reports examining the differences in whether or not they perform each physical examination (alternatively, whether or not they have records), and this would be worth exploring. For example, rectal examination does not contribute much to the diagnostic accuracy of appendicitis [26]. However, Rusnak et al. reported fewer diagnostic errors for appendicitis cases with documented rectal examination [27]. This suggests that the physician's attitude toward physical examination, rather than a direct effect of physical examination, may be related to the accuracy of appendicitis diagnosis.”

Comment 4: Twenty variables were included in the multivariate regression analysis. It is unclear whether interactions were included as well. I would suggest a statistician to evaluate the analysis. 

Response 4: As the reviewer suggested, we revised the multivariate regression model. The new model included following 12 items: 7 items were selected based on the existing literature: female sex, constipation, no pain at presentation, presence of comorbidities, pelvic appendicitis, and age >60 years; and 5 items were selected that considered clinically important: CRP ≥10 mg/L, diarrhea, past history of appendicitis, a consultation >6 h after symptom onset, and physician training (generalist or non-generalist). Regarding the interaction terms, we did not include any interaction term in the model since we assumed that there was no potential factor that affected the diagnostic accuracy of physicians’ specialty. According to the changes, we revised the methods and results sections.

(P8 L179-L185)

“Multivariate logistic regression analysis for delayed diagnosis (diagnostic error) considered the following factors suggested by existing literature: female sex, constipation, no pain at presentation, presence of comorbidities, pelvic appendicitis, and age >60 years. In addition, the following factors, which may be relevant to the diagnostic process and diagnostic errors were also included: CRP ≥10 mg/L, diarrhea, no tenderness in the right lower quadrant, history of appendicitis, a consultation >6 h after symptom onset, and physician training (generalist or non-generalist).”

(P9 L189-191)

“The sample size was targeted at approximately 1000 cases, based on a predicted 20% incidence of delayed diagnosis of acute appendicitis and a planned 12-item multivariate analysis.”

(P10 L221-226)

“On multivariate analysis, the following factors were associated with delayed diagnosis (Table 2): female sex (odds ratio [OR]: 1.95; 95% confidence interval [CI]: 1.21–3.16), no tenderness in the right lower abdominal region (OR: 7.32; 95% CI: 3.45–16.2), diarrhea (OR: 1.91; 95% CI:1.09–3.34), a consultation of >6 h after symptom onset (OR: 2.43; 95% CI: 1.16–5.37), and consultation with a non-generalist (OR: 16.8; 95% CI: 3.19–315).”

■Reviewer2

Comment 1: Introduction: The authors provide adequate comments on the delayed diagnosis of appendicitis. However, there is a lack of information regarding the association between generalists and physical examination, and between appendicitis and physical examination. It is unclear contextually why the authors hypothesized that generalists (or physical examination) would lead to a lower delayed diagnosis of appendicitis. The authors should add information with references in the literature. 

Response 1: Thank you for your constructive comments. As the reviewer suggested, we elaborated the references to clarify the background of this study.

(P3 L59-P4 L86)

In Japan, specialty training has historically been emphasized; therefore, specialists and generalists provide assessment and treatment in primary and emergency care [20, 21]. A multicenter study in Japan regarding the delayed diagnosis of lower gastrointestinal perforation highlighted the possible effect of physician training on diagnostic accuracy for acute abdomen, with a lower rate of delayed diagnosis identified for generalists (primary care and emergency physicians) than for non-generalists [21]. However, the study did not address in detail the factors or processes that led generalists to make fewer diagnostic errors for lower gastrointestinal perforation [21]. Acute appendicitis, like lower gastrointestinal perforation, is an acute abdominal condition with a similar clinical presentation. Therefore, one would expect a lower rate of diagnostic delay in acute appendicitis by generalists; however, no studies have reported this.

Since there is no single clinically significant physical finding that is conclusive for the diagnosis of appendicitis [22] and there is no definitive laboratory test, [23] clinical judgment based on history taking and physical examination is key in diagnosing acute appendicitis. A common pitfall in the diagnosis of appendicitis is including or excluding the diagnosis based on a single sign or symptom [23]. Reports of physical findings made by attending physicians on general medicine inpatients leading to important changes in clinical management [24] and reports that inadequate physical examination causes preventable medical errors [25] suggest that improved accuracy of physical examination findings is important for an excellent diagnostic process for acute appendicitis. While there have been studies on the degree of agreement among physicians when they take the same physical examination, there have been few reports examining the differences in whether or not they perform each physical examination (alternatively, whether or not they have records), and this would be worth exploring. For example, rectal examination does not contribute much to the diagnostic accuracy of appendicitis [26]. However, Rusnak et al. reported fewer diagnostic errors for appendicitis cases with documented rectal examination [27]. This suggests that the physician's attitude toward physical examination, rather than a direct effect of physical examination, may be related to the accuracy of appendicitis diagnosis.

Comment 2: The objectives stated in the Introduction, and the conclusions of this study seem to be disconnected. The authors should conclude with discussions of the study's findings with objective information already known and a clear answer to the objectives and hypotheses. 

Response 2: As the reviewer suggested, we revised the conclusion.

(P17 L345-351)

In patients who were diagnosed with appendicitis by the CT findings, the delayed diagnosis occurred in around one-fourth. Consultation by generalists at the index visit seemed to be associated with the lower occurrence of delayed diagnosis. Besides, this study also suggested that generalists may perform more variety of physical examination to detect peritoneal irritation, which can be associated with less delayed diagnosis compared to the other specialty. Future studies are warranted to clarify the hypothesis by adjusting several confounding factors such as patient complexity, consultation environment.

Comment 3: Method：Informed consent should not be waived for a retrospective study. At least comprehensive consent or opt-out should be implemented and mentioned in the main text.

Response 3: Since we used opt-out method, we revised the sentence related to informed consent.

(P5 L109-110)

“An opt-out method was used so that patients could refuse to participate in the study.”

Comment 4: Please clarify the symptoms of appendicitis or CT findings used for inclusion, objectively and specifically, with references if possible. Additionally, please clarify how the accuracy of the review of medical records was ensured, e.g., how many reviewers were responsible for the review. 

Response 4: We included only patients whose CT reports described appendicitis as the diagnosis. We revised method section and discussion section as follows:

(P6 L120-121)

“We only included patients whose radiology reports described appendicitis as the diagnosis by reviewing the medical records of eligible patients.”

(P6 L130-131)

“In this study, the diagnosis of acute appendicitis was defined as that acute appendicitis was described as the diagnosis in the radiology CT report.”

(P16 L310-314)

“Second, objective criteria were used for CT findings and diagnostic delay but were evaluated by a single reviewer. Third, whether at the clinic or hospital, the number of physicians and outpatients and the differences among patients may have been poorly adjusted for by the wide variety of confounding factors in different consultation settings.”

Comment 5: The inclusion criteria are not fully explained. 

Response 5: We revised the sentence related with the inclusion criteria.

(P 6 L113-127)

“We used two datasets from Showa University Koto Toyosu Hospital to extract the cases of appendicitis: one was Japanese Healthcare Insurance system database, and the other was radiology reports of CT. In this study, we extracted the medical records with the disease code of appendicitis in insurance system database or with the diagnosis of appendicitis was documented in the CT report. Eligible patients were ≥ 15 years of age who came directly or were referred from other clinics to Showa University Koto Toyosu Hospital between April 1, 2014, and March 31, 2021. In the referred groups, simple blood tests such as complete blood count was conducted in some patients. We only included patients whose radiology reports described appendicitis as the diagnosis by reviewing the medical records of eligible patients. Only the first event was included for patients with repeat visits for possible acute appendicitis to avoid the confounding effects of prior notes on the diagnostic process. Excluded were cases in which the final diagnosis was not acute appendicitis, the onset of appendicitis was during the period of hospitalization, the diagnosis was stump appendicitis, and prior hospitalization for acute appendicitis at another hospital, with patients transferred to our hospital for treatment.”

Comment 6: When was the CT performed, and when did the patients have the reviewed symptoms? 

Response 6: We did not collect the time that CT was performed in each patient. In the majority of the timely diagnosis groups, CT was performed within hours of the day of the visit to the hospital. Symptoms of patients referred from clinics to the hospital were mainly collected from referral letter, but additional information were also collected from the chart at the first visit to the hospital.

We have added the manuscript in method as follows:

(P7 L144-149)

Symptoms and physical findings were confirmed using physicians' and nurses' medical records at Showa University Koto Toyosu Hospital and other records, such as scanned referral letters, medical questionnaires, and imaging data. Information on patients visited by referral was collected first from the information in the patient referral document, and the missing information was made up with the information at Showa University Koto Toyosu Hospital.

(P8 L171-173)

In other words, CT was performed at the first visit to Showa University Koto Toyosu Hospital in almost all of the timely diagnosis group.

Comment 7: Please explain how missing data is managed. Mainly, how the authors managed physical findings and symptoms being not recorded is essential. Please clarify whether the authors counted them as none or missing. Additionally, please describe in the limitation how this may affect the results of this study. 

Response 7: We treated no description about each symptom and physical finding as missing data. To clarify the effect of the missing data, we added the sensitivity analysis by using multiple imputation method, which did not change the main results at least about specialty on the diagnostic errors.

(P8 L186-P9 L188)

“We also conducted sensitivity analysis for multivariate logistic regression analysis by using multiple imputation method to handle missing data. We used the “mice” package of R for multiple imputation with 20 imputations.”

(P10 L227-P11 L238)

“Regarding the variables included in the multivariate analysis, at least one missing data existed in 305/763 cases (40.0%): 253 in constipation (33.2%), 184 in diarrhea (24.1%), 61 in physician training (8.0%), 22 in no tenderness in the right lower abdominal region (2.9%), 8 in no pain at presentation (1.1%), 7 in pelvic appendicitis (0.9%), 5 in a consultation >6 h after symptom onset (0.7%), and 2 in CRP ≥10 mg/L (0.3%). The prevalence of at least one missing data was not different between the groups of delayed diagnosis (218/563, 38.7%) and timely diagnosis (87/200, 43.5%) (P=0.24). In the multivariate analysis with multiple imputations, no tenderness in the right lower quadrant (OR: 5.93; 95% CI: 3.11-11.3), diarrhea (OR: 1.96; 95% CI: 1.21-3.18), consultation of >6 h after symptom onset (OR 2.38; 95% CI: 1.27-4.45), and consultation by a non-generalist (OR 7.00; 95% CI: 1.97-24.9) remained significant risk factors for delayed diagnosis.”

Comment 8: How were the collected items selected? Did the authors review the literature?

Response 8: We collected items that are associated with diagnostic errors in appendicitis based on the prior literatures. We also collected clinically relevant items that may be used to determine if the patient has appendicitis, such as items of ALVARADO score, inflammatory markers, and history of appendicitis.

Comment 9: Please explain whether CT with or without contrast enhancement was considered and how and why it was determined. 

Response 9: It is at the discretion of each clinician regarding the use of contrast media. The difference in sensitivity due to the use of contrast media is very small, about 1% [47], and we did not include it in the collection as we judged that the difference was unlikely to be associated with a delayed diagnosis of appendicitis in this study.

We have added the following to the limitation section in response to the reviewer’s suggestion as follow:

(P17 L336-341)

In addition, the use of contrast media in CT imaging is left to the discretion of each clinician, and we did not investigate the association with delayed diagnosis. The difference in sensitivity of CT for acute appendicitis due to the use of contrast media is very small, about 1%, and we did not collect information in this study because we judged that the association with delayed diagnosis of appendicitis is very low [47].

Comment 10: Shouldn’t the diagnostic accuracies in other modalities be discussed?

Response 10: We added the discussion about the diagnostic accuracies in other modalities.

(P16 L321-P17 L335)

Fifth, only cases of acute appendicitis confirmed by abdominal CT were included; therefore, cases diagnosed by sonography, which included pregnant women, were excluded. CT scan is a highly sensitive and specific test for acute appendicitis. Of course, clinical diagnosis and ultrasonography are also important, but clinical judgment and ultrasonography are often not documented, and there are differences in accuracy among examiners. Since this study focused on "diagnostic error," we included cases with confirmed appendicitis by CT as criteria, emphasizing information for which more objective data remained. Japan has many CT machines with good accessibility [46], and CT is commonly used to make decisions in adult cases. However, CT also has problems with radiation exposure and variation in accessibility. It should be noted that this study was conducted in central Japan, where access to CT is good in both clinics and hospitals. Sixth, this study focuses on the outcome of delayed diagnosis in cases in which a CT scan confirms the diagnosis of acute appendicitis. The use of other modalities, prognosis, length of hospital stay, and complications was not investigated; therefore, it is impossible to determine whether early diagnosis by CT is the optimal diagnostic process.

Comment 11: I think that performing CT and referring patients for CT should be different. What is the meaning to analyze patients from clinics without CT and hospitals with CT as the same population? Please clarify in the main text. C

Response 11: Since we believe that the most vital part for the timely diagnosis of acute appendicitis is suspecting the disease based on the history taking and physical examination, which can be conducted irrespective to the location. Therefore, we included patients who directly visited our hospital and also patients who were referred from other clinics in this study. We addressed this issue in the last paragraph of introduction section:

(P5 L91-95)

The most vital part for the timely diagnosis of acute appendicitis is suspecting the disease based on the history taking and physical examination, which can be conducted irrespective to the location. Therefore, we included patients who directly visited our hospital and also patients who were referred from other clinics in this study.

Comment 12: The other biases that affect differences in the location of the first consultation seem not to be adjusted sufficiently. 

Response 12: We have changed the manuscript in discussion as follows:

(P16 L308-314)

“First, this is a single-center study that used a retrospective design. Therefore, selection and information bias cannot be ruled out, nor external validity examined. Second, objective criteria were used for CT findings and diagnostic delay but were evaluated by a single reviewer. Third, whether at the clinic or hospital, the number of physicians and outpatients and the differences among patients may have been poorly adjusted for by the wide variety of confounding factors in different consultation settings.”

Comment 13: Although the duration from onset (6 hours) and age (60 years) were analyzed as categorical variables, they must be analyzed as continuous variables. If the authors analyze them as categorical variables, they should explain how they set the cut-offs. 

Response: We analyzed age as continuous variable in the revised manuscript. The duration between the onset and the index visit were not recorded as continuous variable, therefore, we analyzed that as categorical variable. The cut-off value of 6 hours was set based on prior literature.

Patient factors (age, sex, the presence of underlying disease, consultation >6 h after symptom onset [28], and no history of appendicitis),

Comment 14: The bias for delayed diagnosis due to the clinical settings would not be adjusted unless the duration from the onset in detail is included in the covariate items. 

Response 14: Since we were not able to adjust the duration from the onset in detail, we added a description in discussion as follows:

(P16 L308-314)

“First, this is a single-center study that used a retrospective design. Therefore, selection and information bias cannot be ruled out, nor external validity examined. Second, objective criteria were used for CT findings and diagnostic delay but were evaluated by a single reviewer. Third, whether at the clinic or hospital, the number of physicians and outpatients and the differences among patients may have been poorly adjusted for by the wide variety of confounding factors in different consultation settings.”

Comment 15: The differences in the departments may reflect the differences in the visiting patients. Depending on the role of each department, the patients’ backgrounds should be different, such as a patient with a difficult diagnosis visiting a specialist or a first-time patient visiting a generalist. How did the authors address such biases? 

Response: We have added the following to the limitation section in response to the reviewer’s suggestion as follow:

(P16 L311-314)

“Third, whether at the clinic or hospital, the number of physicians and outpatients and the differences among patients may have been poorly adjusted for by the wide variety of confounding factors in different consultation settings.”

Comment 16: What is the definition of the generalist in this study? Please clarify whether they are board-certified physicians or simply belong to a general medicine department.

Response 16: We have changed the manuscript in method as follows:

(P7 L150-153)

Board-certified primary care and emergency physicians and the trainees were classified as generalists, as previously described [21]; other board-certified physicians and the trainees were classified as non-generalists, with subclassifications of gastroenterologists and non-gastroenterologists.

Comment 17: Because the primary outcome of this study is a delayed diagnosis, potential confounding factors such as type of department, number of medical doctors, number of beds, and number of outpatients in each department should be clarified and adjusted in the multivariate analysis.

Response 17: The personnel turnover at Showa University Koto Toyosu Hospital is rapid, and it was difficult to identify potential confounding factors and adjust for them in a multivariate fashion. In response to the reviewer’s suggestion, we have changed in limitation as follows:

(P16 L308-314)

“First, this is a single-center study that used a retrospective design. Therefore, selection and information bias cannot be ruled out, nor external validity examined. Second, objective criteria were used for CT findings and diagnostic delay but were evaluated by a single reviewer. Third, whether at the clinic or hospital, the number of physicians and outpatients and the differences among patients may have been poorly adjusted for by the wide variety of confounding factors in different consultation settings.”

Comment 18: The first paragraph of the Exploration analysis should be included in the background.

Response 18: We have deleted the relevant part in method, and added a description in introduction as follows:

(P4 L70-86)

“Since there is no single clinically significant physical finding that is conclusive for the diagnosis of appendicitis [22] and there is no definitive laboratory test, [23] clinical judgment based on history taking and physical examination is key in diagnosing acute appendicitis. A common pitfall in the diagnosis of appendicitis is including or excluding the diagnosis based on a single sign or symptom [23]. Reports of physical findings made by attending physicians on general medicine inpatients leading to important changes in clinical management [24] and reports that inadequate physical examination causes preventable medical errors [25] suggest that improved accuracy of physical examination findings is important for an excellent diagnostic process for acute appendicitis. While there have been studies on the degree of agreement among physicians when they take the same physical examination, there have been few reports examining the differences in whether or not they perform each physical examination (alternatively, whether or not they have records), and this would be worth exploring. For example, rectal examination does not contribute much to the diagnostic accuracy of appendicitis [26]. However, Rusnak et al. reported fewer diagnostic errors for appendicitis cases with documented rectal examination [27]. This suggests that the physician's attitude toward physical examination, rather than a direct effect of physical examination, may be related to the accuracy of appendicitis diagnosis.”

Comment 19: The second paragraph of the Exploration analysis should be described in Data collection and definition of variables. 

Response 19: We have deleted the relevant part, and modified the content and appended it in Data collection and definition of variables as follows:

(P7 L156-P8 L165)

For the exploratory analyses, we included only patients who presented directly to Showa University Koto Toyosu Hospital as their first medical institution after symptom onset. Detailed information was collected via a review of medical records. As a result, the following information on physical findings was extracted: right lower abdominal tenderness, tenderness at McBurney's point, tenderness at Lanz's point, cough, percussion, heel drop, psoas, and obturator test. Right lower abdominal tenderness, tenderness at McBurney's point, and tenderness at Lanz's point were physical findings related to the right lower abdomen; the cough [32‒37], percussion [32‒34, 36, 37], and heel drop tests [38‒40] were physical tests aimed at detecting minor perineal irritation. The psoas and obturator tests were classified as tests that could detect pelvic appendicitis [37, 41].

Comment 20: The number of patients should not be mentioned as 247 in the method. This should be mentioned in the results. 

Response 20: We have deleted the relevant part.

Comment 21: Discussion. This study design does not adequately adjust for differences between patients in specialist and generalist departments. It is impossible to determine whether the difference between the two groups is truly significant unless the biases such as patient characteristics, symptom course, and other conditions are adjusted. 

Response 21: We agree with the reviewer. We treated the issue as limitation.

(P16 L308-314)

First, this is a single-center study that used a retrospective design. Therefore, selection and information bias cannot be ruled out, nor external validity examined. Second, objective criteria were used for CT findings and diagnostic delay but were evaluated by a single reviewer. Third, whether at the clinic or hospital, the number of physicians and outpatients and the differences among patients may have been poorly adjusted for by the wide variety of confounding factors in different consultation settings.

Comment 22: This is somewhat of a leap in logic. While it is true that the generalist performed CT earlier, this study's results cannot determine if this was the correct diagnostic practice.

Response 22: This study did not examine the use of other modalities, prognosis, length of hospital stays, complications. Therefore, we think that the results of this study cannot be used to determine whether early diagnosis by CT is the correct method of diagnosis, as the reviewer have pointed out.

(P16 L321-P17 L336)

Fifth, only cases of acute appendicitis confirmed by abdominal CT were included; therefore, cases diagnosed by sonography, which included pregnant women, were excluded. CT scan is a highly sensitive and specific test for acute appendicitis. Of course, clinical diagnosis and ultrasonography are also important, but clinical judgment and ultrasonography are often not documented, and there are differences in accuracy among examiners. Since this study focused on "diagnostic error," we included cases with confirmed appendicitis by CT as criteria, emphasizing information for which more objective data remained. Japan has many CT machines with good accessibility [46], and CT is commonly used to make decisions in adult cases. However, CT also has problems with radiation exposure and variation in accessibility. It should be noted that this study was conducted in central Japan, where access to CT is good in both clinics and hospitals. Sixth, this study focuses on the outcome of delayed diagnosis in cases in which a CT scan confirms the diagnosis of acute appendicitis. The use of other modalities, prognosis, length of hospital stay, and complications was not investigated; therefore, it is impossible to determine whether early diagnosis by CT is the optimal diagnostic process. Also, it is possible that generalist are performing CT imaging more frequently.

Comment 23: Since there are many factors to be considered, such as patients for whom CT was not performed, patients diagnosed by other modalities, and differences in patients according to the department, it would be preferable to use a non-judgmental expression here. 

Response 23: We have changed the manuscript reｌevant part as follows:

(P15 L291-299)

“A key finding in our study was that generalists were more likely than non-generalists to perform various physical tests. They did not limit their assessments to the right lower abdomen or rely on tenderness in the right lower abdomen for diagnosis. Specifically, generalists may include tests for minor peritonitis (cough, percussion, and heel drop tests) [32‒40] and appendicitis in the pelvic region (psoas and obturator test) to improve their diagnostic accuracy. Our finding that generalists may have a more complete diagnostic process than non-generalists was consistent with a previous study which reported that compared with non-generalists, generalist physicians have a lower risk of delayed diagnosis for lower gastrointestinal perforation, another acute abdomen condition [21].”

Comment 24: Additionally, the physical examination was only analyzed in univariate analysis, and it is difficult to adjust adequately with other factors considering the small sample size. I believe this is a critical flaw in this study to emphasize the superiority of the generalists' physical examination.

Response 24: We considered that to be a limitation of this study.

(P16 L314-321)

Fourth, the exploratory analysis regarding physical examination was univariate only and the sample size was small, making it difficult to adjust for other factors fully. Moreover, although the record of physical findings was checked because of the design of this study, it is impossible to know if the physician took a particular physical finding and did not document it in the medical record. However, the fact that a physical examination finding is documented in the medical record indicates that the physician thought the finding was important to the diagnosis. Thus, the results suggest that the generalist was interested in a more detailed search for intra-abdominal disease.

Comment 25: I cannot understand the relation of the 4th paragraph to the results or the conclusions of this study. 

Response 25: As the reviewer mentioned, we thought that the part you pointed out was not appropriate as a single paragraph. We have made the following changes and added them to the third paragraph

(P15 L299-304)

In recent years, it has been noted that interest in physical examination has been waning with advances in diagnostic imaging [43]. However, general medicine departments in Japan provide an effective clinical situation for educating residents about physical examination [44, 45]. These backgrounds may be related to the present results, in which generalists recorded a greater variety of physical findings.

Comment 26: Conclusion. Please revise the conclusion based on the hypothesis, objectives, findings, and appropriate discussions.

Response 26: We have changed the manuscript in conclusion as follows:

(P17 L345-351)

In patients who were diagnosed with appendicitis by the CT findings, the delayed diagnosis occurred in around one-fourth. Consultation by generalists at the index visit seemed to be associated with the lower occurrence of delayed diagnosis. Besides, this study also suggested that generalists may perform more variety of physical examination to detect peritoneal irritation, which can be associated with less delayed diagnosis compared to the other specialty. Future studies are warranted to clarify the hypothesis by adjusting several confounding factors such as patient complexity, consultation environment

Comment 27: “Second, we could not verify whether physical findings, such as peritoneal irritation or pelvic inflammation, could prevent delayed diagnosis owing to the retrospective study design.” This section is not the conclusion that can be drawn from the findings of this study.

Response 27: We have removed the relevant section.

Comment 28: “Our findings will need validation in future multicenter and prospective studies.” I do not think it is necessary as a conclusion of this study. " 

Response 28: We have removed the relevant section.

---

## [Editor Report · Decision Letter 1]

7 Sep 2022

PONE-D-22-16651R1Factors associated with delayed diagnosis of appendicitis in adults: a single-center, retrospective, observational studyPLOS ONE

Dear Dr. Harada,

Thank you for submitting your manuscript to PLOS ONE. After careful consideration, we feel that it has merit but does not fully meet PLOS ONE’s publication criteria as it currently stands. Therefore, we invite you to submit a revised version of the manuscript that addresses the points raised during the review process.

We look forward to receiving your revised manuscript.

Kind regards,

Kenneth A Michelson, MD MPH

Academic Editor

PLOS ONE

Additional Editor Comments:

The authors state that delayed diagnosis rates have only been reported as negative appendectomy rates. This is categorically not true - there are several previous studies that report delayed diagnosis rates as rates of missed diagnosis on an initial visit, generally < 5%. Please provide some comparative commentary on why your delay rates are so different. Is this a difference in health systems?
---

## [Author Response · Author response to Decision Letter 1]

26 Sep 2022

Thank you very much for reviewing our manuscript and offering valuable advice.

We appreciate the invaluable comments that the reviewers and editors provided, which we are confident have helped us improve the manuscript. We have provided point-by-point responses to each of the comments and describe the related revisions below. We have indicated changes via red characters in the revised manuscript.

Thank you again for your thorough and supportive peer review. 

On behalf of the co-authors,

Sincerely,

Taku Harada

Division of General Medicine, Showa University Koto Toyosu Hospital, 

5-1-38 Toyosu Koto-ku, Tokyo, 135-8577, Japan

hrdtaku@gmail.com

■Additional Editor Comments:

The authors state that delayed diagnosis rates have only been reported as negative appendectomy rates. This is categorically not true - there are several previous studies that report delayed diagnosis rates as rates of missed diagnosis on an initial visit, generally < 5%. Please provide some comparative commentary on why your delay rates are so different. Is this a difference in health systems?

Response:

Thank you for pointing out this critical issue.

The frequency of diagnostic errors in appendicitis is vast, we think it is influenced by differences in healthcare systems and definitions of diagnostic errors as you mentioned.

Also, a pediatric study, similar to ours, that defined diagnostic delay based on the diagnosis at the first consult reported 66% diagnostic delay (JAMA Netw Open. 2021;4(8):e2122248).

In response to your comment, we have added a description in discussion part as follows:

(P15 L287-93)

On the other hand, although a pediatric study, Michelson et al. in their multicenter case-control study of diagnostic errors in appendicitis, in which the definition of diagnostic delay was strictly defined as failure to diagnose at the first consult as in our study, reported that 63.0% of patients had a diagnostic delay [42]. Despite similar definitions of diagnostic delay, we believe that this large difference in the frequency of diagnostic errors in appendicitis is also influenced by situativity, including patient age and differences in health care systems [43].

---

## [Editor Report · Decision Letter 2]

28 Sep 2022

PONE-D-22-16651R2Factors associated with delayed diagnosis of appendicitis in adults: a single-center, retrospective, observational studyPLOS ONE

Dear Dr. Harada,

Thank you for submitting your manuscript to PLOS ONE. After careful consideration, we feel that it has merit but does not fully meet PLOS ONE’s publication criteria as it currently stands. Therefore, we invite you to submit a revised version of the manuscript that addresses the points raised during the review process.

We look forward to receiving your revised manuscript.

Kind regards,

Kenneth A Michelson, MD MPH

Academic Editor

PLOS ONE

Journal Requirements:

Additional Editor Comments:

I am happy with the changes and how you have contextualized the findings. The Michelson study showed that 63% of patients who had a revisit had a delay in diagnosis. Most patients do not have revisits. So, 63% of the small fraction of patients with a revisit leading to an appendicitis diagnosis had a delay. Overall, delay rates are still < 5% among all children diagnosed with appendicitis. As an example, Mahajan et al in JAMA Network Open and Goyal et al in Academic Emergency Medicine show low delay rates. Please modify the text to incorporate the generally low delay rates published in contemporary cohorts.
---

## [Author Response · Author response to Decision Letter 2]

4 Oct 2022

Thank you very much for reviewing our manuscript and offering valuable advice.

We appreciate the invaluable comments that the editors provided, which we are confident have helped us improve the manuscript. We have provided point-by-point responses to each of the comments and describe the related revisions below. 

We also submitted it to a proofreader who corrected the wording to make it more appropriate

We have indicated changes via red characters in the revised manuscript.

Thank you again for your thorough and supportive peer review. 

On behalf of the co-authors,

Sincerely,

Taku Harada

Division of General Medicine, Showa University Koto Toyosu Hospital, 

5-1-38 Toyosu Koto-ku, Tokyo, 135-8577, Japan

hrdtaku@gmail.com

■Journal Requirements:

Response:

Thank you for pointing out the important issue.

We apologize for the duplication of No. 14 and No. 27 in the references during the Revise process.

Reference No.27 has been deleted.

Reference No. 44 has been corrected.

Reference No.41 has been newly added.

■Additional Editor Comments:

I am happy with the changes and how you have contextualized the findings. The Michelson study showed that 63% of patients who had a revisit had a delay in diagnosis. Most patients do not have revisits. So, 63% of the small fraction of patients with a revisit leading to an appendicitis diagnosis had a delay. Overall, delay rates are still < 5% among all children diagnosed with appendicitis. As an example, Mahajan et al in JAMA Network Open and Goyal et al in Academic Emergency Medicine show low delay rates. Please modify the text to incorporate the generally low delay rates published in contemporary cohorts.

Response:

Thank you for pointing out this critical issue.

Additionally, we thanked for the educational review, including the literatures.

In response to your suggestion, we have added a description in discussion part as follows:

[P15 L283-292]

On the other hand, a recent large cohort study in pediatrics reported a delay rate of <5% in the diagnosis of appendicitis [8, 41]. Michelson. et al. reported a 63% delay in appendicitis diagnosis in a study that included only patients in whom a revisit led to the appendicitis diagnosis [42]. This large difference in the frequency of diagnostic errors for appendicitis may be influenced by differences in the definition of diagnostic error, as well as differences in context, including clinical practice and differences in health care systems [43]. We believe that the frequency of delayed diagnosis in our study was higher than the rate reported in previous studies because we used strict the criteria for timely diagnosis to reduce bias, including only cases with a correct diagnosis made at the time of an initial visit as timely diagnosis.

---

## [Editor Report · Decision Letter 3]

7 Oct 2022

Factors associated with delayed diagnosis of appendicitis in adults: a single-center, retrospective, observational study

PONE-D-22-16651R3

Dear Dr. Harada,

We’re pleased to inform you that your manuscript has been judged scientifically suitable for publication and will be formally accepted for publication once it meets all outstanding technical requirements.

Kind regards,

Kenneth A Michelson, MD MPH

Academic Editor

PLOS ONE
---

## [Editor Report · Acceptance letter]

12 Oct 2022

PONE-D-22-16651R3 

Factors associated with delayed diagnosis of appendicitis in adults: a single-center, retrospective, observational study 

Dear Dr. Harada:

I'm pleased to inform you that your manuscript has been deemed suitable for publication in PLOS ONE. Congratulations! Your manuscript is now with our production department. 

Kind regards, 

on behalf of

Dr. Kenneth A Michelson 

Academic Editor

PLOS ONE